# Fabrication of Flexible Poly(*m*-aminophenol)/Vanadium Pentoxide/Graphene Ternary Nanocomposite Film as a Positive Electrode for Solid-State Asymmetric Supercapacitors

**DOI:** 10.3390/nano13040642

**Published:** 2023-02-06

**Authors:** SK Safdar Hossain, Baban Dey, Syed Sadiq Ali, Arup Choudhury

**Affiliations:** 1Department of Chemical Engineering, College of Engineering, King Faisal University, P.O. Box 380, Al-Ahsa 31982, Saudi Arabia; 2Department of Chemical Engineering, Birla Institute of Technology, Ranchi 835215, India

**Keywords:** V_2_O_5_ nanoparticles, poly(*m*-aminophenol), conductive binder, asymmetric cell, energy density

## Abstract

In this study, poly(*m*-aminophenol) (PmAP) has been investigated as a multi-functional conductive supercapacitor binder to replace the conventional non-conductive binder, namely, poly(vinylene difluoride) (PVDF). The kye benefits of using PmAP are that it is easily soluble in common organic solvent and has good film-forming properties, and also its chemical functionalities can be involved in pseudocapacitive reactions to boost the capacitance performance of the electrode. A new ternary nanocomposite film based on vanadium pentoxide (V_2_O_5_), amino-functionalized graphene (amino-FG) and PmAP was fabricated via hydrothermal growth of V_2_O_5_ nanoparticles on graphene surfaces and then blending with PmAP/DMSO and solution casting. The electrochemical performances of V_2_O_5_/amino-FG/PmAP nanocomposite were evaluated in two different electrolytes, such as KCl and Li_2_SO_4_, and compared with those of V_2_O_5_/amino-FG nanocomposite with PVDF binder. The cyclic voltametric (CV) results of the V_2_O_5_/amino-FG/PmAP nanocomposite exhibited strong pseudocapacitive responses from the V_2_O_5_ and PmAP phases, while the faradaic redox reactions on the V_2_O_5_/amino-FG/PVDF electrode were suppressed by the inferior conductivity of the PVDF. The V_2_O_5_/amino-FG/PmAP electrode delivered a 5-fold greater specific capacitance than the V_2_O_5_/amino-FG/PVDF electrode. Solid-state asymmetric supercapacitors (ASCs) were assembled with V_2_O_5_/amino-FG/PmAP film as a positive electrode, and their electrochemical properties were examined in both KCl and Li_2_SO_4_ electrolytes. Although the KCl electrolyte-based ASC has greater specific capacitance, the Li_2_SO_4_ electrolyte-based ASC delivers a higher energy density of 51.6 Wh/kg and superior cycling stability.

## 1. Introduction

Our daily lives are increasingly dependent on energy, resulting in an increase in global energy demand. The over-exploitation of fossil fuels and their irreversible adverse effects on the environment and the climate led to the promotion of clean energy resources. Electrochemical energy conversion and storage (EECS) technologies, including rechargeable batteries, fuel cells, and supercapacitors, can replace coal-based power supplies. As a next-generation energy storage device, supercapacitors have been proven to be the most reliable and effective alternative to conventional batteries since batteries demonstrate slow power delivery and uptake, a short cycle-life, and a less eco-friendly profile, while supercapacitors offer high power density, a fast delivery rate, high safety performance, and a long-life cycle (up to 10^4^) [1,2]. For portable and flexible electronics, such as tablets, laptops, and foldable phones, flexible solid-state supercapacitors can serve as a real-time power backup system and micropower source instead of batteries for these flexible smart electronics [3,4,5,6]. Yet, state-of-the-art supercapacitors offer a limited energy storage density a few orders of magnitude less than conventional batteries, which prevents their widespread use in the energy-storage industries [7,8]. Generally, there are two mechanisms by which supercapacitors store electrical energy: electronic double layer capacitance (EDLC) and pseudocapacitance. For EDLC, the charge storage process occurs through the formation of electrical double-layers by means of the adsorption/desorption of electrolyte ions onto the active electrode surface, while the pseudocapacitive materials store charges via the faradaic charge storage process [5,6].

Generally, EDLC-electrodes are made of porous carbon, including activated carbon, mesoporous carbon, and graphitic nanocarbons [9,10]. Meanwhile, graphene—a thin, 2D, carbon sheet—has received huge interest as an advanced EDLC active material due to its superior electrical conductivity, exceptionally high charge carrier mobility (~15,000 cm^2^/V/s), large theoretical specific surface area (up to 2675 m^2^/g), superior theoretical gravimetric capacitance (500 F/g), strong mechanical strength (~1 TPa), and low cost [11,12]. However, the low capacitance of graphene-based electrode materials still remains a fundamental challenge for their widespread commercial use in supercapacitors. The strong agglomeration tendency of graphene sheets significantly reduces their effective surface area and ion-diffusion rate [13]. Several studies have demonstrated that carbon materials combined with pseudocapacitive materials, such as metal oxides [14,15,16], metal hydroxide [17,18], metal sulfides [19,20], metal-organic framework [21,22,23], and conducting polymers [24,25], can improve the capacitance of carbon-based supercapacitors because of their synergistic effects.

Among the several transition-metal-oxides-based pseudocapacitive electrode materials, vanadium pentoxide has gained considerable attention, either as a pseudocapacitive electrode material for supercapacitors or as a high-capacity cathode material for Li-/Na-ion batteries [26,27,28], because of its layered-crystalline structure, rich redox chemistry (V^+3^, V^+4^, and V^+5^), ultra-high theoretical capacitance of 2120 F/g over a voltage window of 1.0 V, and earth-abundant and non-toxic nature [29,30]. In spite of these attractive advantages, the low electrical conductivity and inferior structural stability (in liquid electrolytes) of V_2_O_5_ inhibit the charge–discharge rate and hamper the long-term cycling performance [31]. There has been a lot of research done to enhance the ion-diffusion rate of V_2_O_5_ and subsequently improve its capacitive performance. The synthesis of hybrid materials, by incorporating highly conductive carbon materials or conducting polymers into V_2_O_5_, is the most common approach to overcome those barriers. The specific capacitance of V_2_O_5_ was found to enhanced by 257% [32], 520% [33], 129% [34], 28% [35], and 64% [36] by inclusion of exfoliated graphene, reduced graphene oxide, multi-wall carbon nanotube, N-rich mesoporous carbon, and carbon sphere, respectively. Several researchers have successfully improved the specific capacitance of V_2_O_5_ by hybridization with polyaniline (PANI) [37,38], polypyrrole (PPy) [39,40], polythiophene (PEDOT) [41], and polydiphenylamine [42].

In recent years, the hybridization of an EDLC material with two different pseudocapacitive materials was found to be a highly effective strategy in producing an electrode with enhanced electrochemical properties for supercapacitors. The combination of carbon material (EDLC) with pseudocapacitive materials can not only enhance the capacitance of carbon material, but also boost the chemical stability of the pseudocapacitive components during electrochemical cycling. This is not only because of the synergistic effects between these electrode materials, but also as a result of the enhanced effective surface area, which in turn reduces the diffusion path length. Many research groups have explored various types of ternary hybrid materials, such as graphene/NiO/PPy [43], graphene oxide/Fe_2_O_3_/PPy [44], rGO/MnFe_2_O_4_/PPy [45], graphene/SnO_2_/PEDOT [46], graphene/NiO/poly(aniline-co-m-aminophenol) [47], RGO/ZrO_2_/PPy [48], and graphene/MoO_3_/PPy [49], for supercapacitor applications. Most of the conventional synthetic approaches usually produce powdery active materials, and consequently, nonconductive polymeric additives/conductive agents are required to add to the active materials for fabrication of supercapacitor electrodes, which inescapably reduces the content of active materials, as well as affects their electrical conductivity [50,51]. Conducting polymers as an active component could compensate for the lower electrical conductivity of the electrode material, as well as provide additional pseudocapacitance induced by their functional groups in the backbone chain. However, the inferior solubility of the conducting polymers in most of the organic solvents makes it difficult to prepare free-standing film by solution casting, which limits their use as an active material in supercapacitors. PmAP, a derivative of polyaniline, possesses good electrical conductivity (10^−5^ S/cm), as well as high solubility in organic solvents. The high solubility of PmAP in a common organic medium can allow preparing flexible nanocomposite electrodes via the solution casting approach. In addition, the multiple functional groups in the PmAP backbone chain can act as active redox sites to provide pseudocapacitance and surface wettability, which were observed in our earlier work [23]. These advantages can make PmAP an attractive alternative polymeric binder for commonly used non-conductive binders, namely, PVDV, polytetrafluoroethylene (PTFE), polyvinyl alcohol (PVOH), etc. To the best of our knowledge, the soluble conducting polymers are rarely explored as conductive binders for fabrication of supercapacitor electrodes.

In the present investigation, we have synthesized V_2_O_5_/amino-FG/PmAP nanocomposite film by a two-step process: (i) V_2_O_5_ nanoparticles were initially grown on the surface of amino-FG nanosheets via the hydrothermal method, and (ii) as-prepared V_2_O_5_/amino-FG nanocomposites were dispersed in a PmAP solution and then cast on a glass substrate to form ternary nanocomposite film upon evaporation of the solvent. As far as we know, we are exploring PmAP as binder-cum-pseudocapacitive electrode material for the first time for the possible replacement of commercial PVDF binder. The electrochemical performances of the hydrophilic V_2_O_5_/amino-FG/PmAP nanocomposite film were evaluated and compared in different electrolytes, such as KCl and Li_2_SO_4_. The rich redox chemistry of V_2_O_5_ and the conjugated PmAP backbone can offer pseudocapacitance to boost the specific capacitance of the nanocomposite electrode, while graphene as a conductive component would provide high power density. This bendable nanocomposite film could potentially be used to fabricate flexible supercapacitors for wearable electronic devices. Solid-state ASC cells with different electrolytes were constructed by using ternary nanocomposite film as the positive electrode and activated carbon cloth as the negative electrode.

## 2. Experimental Section

### 2.1. Materials

The aminated graphene (batch: 900551), *m*-aminophenol, V_2_O_5_ powder, potassium chloride (KCl), lithium sulfate (Li_2_SO_4_), nafion solution (5 wt%), ammonium persulfate (APS), and dimethyl sulfoxide (DMSO) were procured from Sigma Aldrich, Mumbai, India. Poly(vinylene difluoride), sulfuric acid (H_2_SO_4_), N-methyl pyrrolidone (NMP), ethanol, ammonium hydroxide (NH_4_OH), ethanol (EtOH), and sodium hydroxide (NaOH) were purchased from Alfa Aesar, Mumbai, India. Activated carbon cloth (Spectracarb 2225) was obtained from Engineered Fibers Technology, LLC, Shelton, CT, USA.

### 2.2. Synthesis of V_2_O_5_/Amino-FG Nanocomposites

The V_2_O_5_/amino-FG nanocomposites were synthesized by the hydrothermal approach. The first step in this synthesis was to disperse 250 mg V_2_O_5_ in 25 mL of deionized (DI) water by mechanical stirring at 70 °C. In the next step, 10 mL of H_2_O_2_ (30%) was added drop-by-drop to the V_2_O_5_ solution under constant stirring, while keeping the solution at 70 °C. The yellow color dispersed solution became orange color solution after completion of the exothermic reaction. Separately, 500 mg of amino-FG powder was dispersed in 25 mL EtOH by ultrasonication before being transferred to a Teflon-lined autoclave. The V_2_O_5_ sol solution was then added to the graphene solution under constant starring for 2 h and then heated at 120 °C for 24 h. The resulting black precipitate was filtered, repeatedly washed with ethanol and water, and finally dried at 60 °C for 24 h.

### 2.3. Preparation of V_2_O_5_/Amino-FG/Poly(m-aminophenol) Nanocomposite Film

The poly(*m*-aminophenol) was synthesized through the chemical oxidation polymerization of *m*-aminophenol monomer by APS in 1 M aqueous NaOH solution at 4 °C [51]. To prepare nanocomposite film, 250 mg PmAP was dissolved in 10 mL DMSO by mechanical stirring at 25 °C. Afterward, 1 g V_2_O_5_/amino-FG powder was added to the PmAP solution and the stirred mixture overnight. The homogeneous solution was then poured on a cleaned petri dish and the DMSO evaporated by heating at 70 °C. The solvent-cast nanocomposite film was thoroughly washed with water to ensure the complete removal of residual solvent molecules and dried at 70 °C overnight. The thickness of the as-prepared V_2_O_5_/amino-FG/PmAP nanocomposite film was determined to be ~420 μm.

### 2.4. Characterizations

IR-absorption and Raman spectra of V_2_O_5_/amino-FG and V_2_O_5_/amino-FG/PmAP nanocomposites were obtained by IRPrestige-21 spectrophotometer and a Renishaw Raman system 3000, respectively. X-ray diffraction (XRD) patterns of as-prepared nanocomposites were recorded on a Rigaku Smart Lab with Cu Kα radiation (40 kV, 40 mA). The elemental compositions of the nanocomposite samples were determined by XPS (MultiLab 2000) and TEM-EDS analyses. The microstructure of the as-prepared nanocomposites were studied by field-emission scanning electron microscopy (FE-SEM, Hitachi, S-4700, Tokyo, Japan) and a transmission electron microscopy (TEM, JEOL JEM 2100-2100F).

### 2.5. Electrochemical Measurements

Electrochemical measurements with 1M KCl and 1M Li_2_SO_4_ were conducted in an Autolab electrochemical workstation. For the CV tests, a three-electrode cell was constituted with a working electrode, a reference electrode (Ag/AgCl), and a counter electrode (platinum wire). The nanocomposite film was initially cut into a circular sample (diameter = 5 mm) and then fixed to the glassy carbon electrode (GCE), using Nafion conductive solution to serve as the working electrode. To use the V_2_O_5_/amino-FG nanocomposite as electrode material, V_2_O_5_/amino-FG nanocomposite powder (80 wt%), super P (5 wt%), and PVDF binder (20 wt%) were mixed in NMP using stirring for 3 h. The working electrode was prepared by coating the slurry on the surface of GCE, followed by drying at 70 °C. PVDF was selected as a binder because of several advantages, such as better capacitive properties, excellent stability, and lower cost than other polymeric binders like PTFE and PVOH. A two-electrode cell was used to carried out the galvanostatic charge–discharge (GCD), where two square pieces of nanocomposite films (dimension = 1 × 1 cm^2^) were employed as working electrodes. The total mass of the hybrid electrode is 3.417 mg. EIS measurements were performed in a frequency region of 0.01–100,000 Hz, using an AC voltage of 5 mV. The solid-state ASC cells were assembled by using V_2_O_5_/amino-FG/PmAP nanocomposite film as a positive electrode and activated carbon cloth as a negative electrode, and their electrochemical performances were evaluated in KCl and Li_2_SO_4_ electrolytes. The polyvinyl alcohol (PVA) + electrolyte membrane was used as electrolyte-cum-separator [52]. The mass ratio of V_2_O_5_/amino-FG/PmAP electrode and activated carbon cloth was calculated to be 0.5823. The specific capacitances (*C_sp_*) of the nanocomposite electrodes were determined from the equation [5]:(1)$$C_{sp}=\frac{I\times \Delta t}{m\times \Delta V}\;$$
where *I*, Δ*t*, *m* and Δ*V* denote the current, discharge time, mass of the active electrode material, and voltage window, respectively. Similarly, the energy density (*ED*) and power density (*PD*) were calculated with the following equations [5]:(2)$$ED=\frac{C_{sp}\times \Delta V^{2}}{2\times 3.6}\;$$
(3)$$PD=\frac{3600\times ED}{\Delta t}\;$$

## 3. Results and Discussion

The FTIR spectra of binary V_2_O_5_/amino-FG nanocomposite and ternary V_2_O_5_/amino-FG/PmAP nanocomposite film are shown in Figure 1a. The FTIR spectrum of V_2_O_5_/amino-FG nanocomposite displays few characteristic absorption peaks at 1616, 1008, 771, and 484 cm^−1^, which refer to the bending vibration of the crystalline water molecule, stretching modes of terminal V=O, V-O-V asymmetrical stretching, and stretching vibration of the O-(V)_3_ bond, respectively [53]. The peak at 1222 cm^−1^ is related to the vibrational modes of the C-O group in the aminated graphene. The broad band in the 3600–2700 cm^−1^ regime denotes O-H stretching. In contrast, the ternary V_2_O_5_/amino-FG/PmAP nanocomposite film displayed some additional IR peaks, along with the absorption bands related to V_2_O_5_. A couple of characteristic peaks at 2920 and 2343 cm^−1^ correspond to stretching of C=C bonds in benzene rings and stretching modes of C=C=N and C=C=O bonds [54]. The C=C stretching mode of the quinoid form appeared at 1611 cm^−1^, while that of the benzenoid form is observed at 1577 cm^−1^. IR spectrum reveals two absorption peaks at 1611 and 1577 cm^−1^ for the quinoid and benzenoid ring stretching vibrations, respectively [55]. The peak at 1144 cm^−1^ refers to C-O-C linkages in the backbone chain [56]. Figure 1b presents the Raman spectra of V_2_O_5_/amino-FG nanocomposite and ternary V_2_O_5_/amino-FG/PmAP nanocomposite films. For the V_2_O_5_/amino-FG nanocomposite, the characteristic peaks in the range frequency range of 200–1000 cm^−1^ are associated with the crystalline V_2_O_5_ phase. The G- and D-band are located at 1571 and 1355 cm^−1^, which refer to the vibration of the sp^2^-carbon lattice in the graphite plane and disordered carbon lattice (sp^3^-carbon), respectively. The peaks at 991, 688, and 526 cm^−1^ are related to stretching modes of terminal V=O, V_2_-O, and V_3_-O bonds, respectively [33]. The peak located at 404 cm^−1^ is due to the V=O bending vibration. As shown in the Raman spectra of V_2_O_5_/amino-FG/PmAP film, in addition to D and G bands, five characteristic bands related to the PmAP phase are observed at 1614, 1486, 1264, 1228, and 1163 cm^−1^, which refer to the stretching absorption of the C=C bond, the stretching mode of the C=N bond, the stretching of the C-N bond in quinoid/polaron forms, the in-plane bending of the C-H bond in benzenoid, and in quinoid units, respectively. The XRD patterns of as-synthesized nanocomposites are shown in Figure 1c. The V_2_O_5_/amino-FG nanocomposite demonstrates characteristic diffraction peaks at 2θ = 6.8°, 26.3°, 31.7°, 34.6°, 46.7°, 50.5°, and 61.4°, which are associated with the (001), (110), (220), (210), (310), (006), and (601) crystal planes of orthorhombic V_2_O_5_ phase (JCPDS file No. 40-1296), respectively. The d-spacing of V_2_O_5_.xH_2_O was calculated to 0.65 nm from the diffraction angle of the (001) plane, using the Bragg equation. In contrast to V_2_O_5_/amino-FG nanocomposite, the diffraction pattern of V_2_O_5_/amino-FG/PmAP nanocomposite shows additional peaks at 2θ = 9.1°, 15.2°, 17.5°, 20.4°, and 23.2° corresponding to the (001), (111), (201), (020), and (200) crystal planes of the PmAP phase, respectively, along with diffraction peaks from the V_2_O_5_ phase. The diffraction peak of the (001) plane of V_2_O_5_ for V_2_O_5_/amino-FG/PmAP nanocomposite was shifted to the lower 2θ value of 6.39° compared to that of V_2_O_5_/amino-FG nanocomposite. The d-spacing of V_2_O_5_.xH_2_O was calculated to 0.691 nm from the diffraction angle of the (001) plane. In the ternary nanocomposite, the enhanced d-spacing of V_2_O_5_.xH_2_O might be attributed the intercalation of PmAP chains between V_2_O_5_ layers, which results in a widening of interplanar spacing of the (001) plane of orthorhombic V_2_O_5_ phase [57].

The surface chemical compositions of V_2_O_5_/amino-FG and V_2_O_5_/amino-FG/PmAP nanocomposite film was evaluated by XPS. The survey spectrum of V_2_O_5_/amino-FG/PmAP nanocomposite film displays four distinct signals for C, N, O, and V elements, indicating their presence in the nanocomposite, while the XPS spectrum of V_2_O_5_/amino-FG powder indicates the existence of C, O, and V elements (Figure 2a). The high-resolution V2p spectrum exhibits V2p_3/2_ and V2p_1/2_ peaks at 513.8 eV and 521.3 eV, respectively (Figure 2b). The O1s spectrum could be deconvoluted into four peaks by Gaussian curve fitting. The peaks at 529.9 eV and 530.7 eV are assigned to the V-O-V and V-O-H bonds in the V_2_O_5_ lattice (Figure 2c) [58]. The small broad peak centered at 533.2 eV is associated with crystal water in V_2_O_5_, indicating the hydrous nature of the vanadium oxide on graphene sheets. The XPS peak at 534.3 eV is attributed to the phenolic oxygen present in PmAP backbone [59,60]. The N1s spectrum can be deconvoluted into two peaks at 399.1 eV and 401.7 eV (Figure 2d), which are related to C-N and C=N^+^- groups in the PmAP backbone.

The microstructures of V_2_O_5_/amino-FG and V_2_O_5_/amino-FG/PmAP nanocomposite films were studied by FE-SEM analysis. The SEM images of V_2_O_5_/amino-FG clearly exhibit that the graphene nanosheets are well wrapped in or covered with tiny V_2_O_5_ nanoparticles. The high-resolution SEM image (Figure 3c) indicates the in situ hydrothermal growth of highly dispersed V_2_O_5_ nanoparticles on the surface of graphene nanosheets. It can be believed that the V_2_O_5_ nanoparticles were sandwiched between graphene sheets. SEM images of the fracture surface of the V_2_O_5_/amino-FG/PmAP nanocomposite film is demonstrated in Figure 3d–f. The microstructure of the fracture surface reveals the structural compactness and the good dispersion of V_2_O_5_-coated graphene nanosheets in the PmAP matrix. Numerous protuberances on the graphene surfaces were observed in high-resolution SEM images, indicating the presence of V_2_O_5_ nanoparticles. This self-assembled interconnected morphology of the nanocomposite would be beneficial for charge transfer between them. The TEM image of the V_2_O_5_/amino-FG/PmAP nanocomposite further demonstrates a consistent dispersion of V_2_O_5_ nanoparticles (average size 10 nm) on graphene surfaces (Figure 3g). The individual V_2_O_5_ nanoparticle clearly exhibits crystal lattice with a lattice spacing of 0.61 nm, corresponding to the d-spacing of (001) lattice planes of the orthorhombic V_2_O_5_ phase. The SAED pattern of the nanocomposite displays well-defined rings with a clear diffraction spot array, indicating the crystalline characteristics of the V_2_O_5_. The diffraction spots in the SEAD pattern can be indexed to the (001), (110), (220), (210), and (310) crystal planes in orthorhombic V_2_O_5_, which is well in agreement with the XRD results. The EDX elemental mapping (Figure 3f) clearly indicates the uniform distribution of carbon, nitrogen, oxygen and vanadium, indicating homogeneous dispersion of V_2_O_5_ nanoparticles throughout the nanocomposite.

The electrochemical properties of as-prepared V_2_O_5_/amino-FG/PmAP nanocomposite films were evaluated in two different electrolytes and compared with those of V_2_O_5_/amino-FG/PVDF nanocomposite. The CV profiles of V_2_O_5_/amino-FG/PmAP nanocomposite-based electrode in 1 M KCl and 1 M Li_2_SO_4_ electrolytes over a potential range of 0–1.0 V at 25 mV/s are demonstrated in Figure 4a. The CV curves of the V_2_O_5_/amino-FG/PmAP electrode obtained in both the electrolytes consist of two pairs of broad oxidation-reduction peaks and are quasi-rectangular in shape. These results suggest that the EDLC from graphene nanosheets combined with the pseudocapacitance from V_2_O_5_ and PmAP makes up the overall capacitance of the nanocomposite electrode. In contrast to the V_2_O_5_/amino-FG/PmAP electrode, the V_2_O_5_/amino-FG/PVDF electrode displays quasi-rectangular CV profiles without redox peaks in either electrolyte, which might be associated with the low conductivity of PVDF and high charge transfer resistance of the V_2_O_5_/amino-FG electrode with PVDF binder compared to V_2_O_5_/amino-FG/PmAP (Figure 5f). The CV profiles of V_2_O_5_/amino-FG/PmAP electrode in KCl electrolyte at different scan rates are displayed in Figure 4b. In the CV curves, two anodic peaks are observed at 0.33 V and 0.72 V, as well as two cathodic peaks at 0.61 V and 0.21 V. When the scan rate is increased from 5 to 100 mV/s, there is a gradual increase in redox peak currents, as well as an increase in the integrated CV area. This could be ascribed to the increased interfacial ionic mobility at higher scan rates. The large number of functional redox-active sites in V_2_O_5_/amino-FG/PmAP nanocomposite makes it suitable for fast reversible faradaic reactions. When the scan rate was increased, there were noticeable shifts in the cathodic and anodic peak potentials towards more positive and negative values, respectively. At higher scan rates, there might be inadequate diffusion of electrolyte ions into nanocomposite matrix as well as restricted charge transfer interactions between electrolytes and electrode surface. Figure 4c illustrates the plots of the redox peak currents against scan rate in logarithmic scale. The slopes of the anodic and cathodic fitting curves are 0.722 and 0.843, respectively. Theoretically, slope value of 0.5 and 1.0 indicates faradaic diffusion process and surface-controlled (pseudocapacitance) process, respectively. Since the slope values of V_2_O_5_/amino-FG/PmAP electrode lie between 0.5 and 1.0, both diffusion and pseudocapacitive behavior affected the electrochemical properties of nanocomposite electrodes. However, their capacitance behavior is dominated by pseudocapacitance process rather than diffusion process, probably because of high ionic mobility of the K^+^ ions (0.68 × 10^−3^ cm^2^/s/V). The electrochemical insertion/desertion of K^+^ ions into the hybrid electrode occurs via the possible reaction: V_2_O_5_ + *n*K^+^ + *n*e^−^ ↔ K_n_V_2_O_5_, where *n* is mole fraction of K^+^ ions. The GCD curves of V_2_O_5_/amino-FG/PmAP electrode in 1 M KCl at different current densities are illustrated in Figure 4d. The non-linearity of the GCD profiles could be attributed to the occurrence of faradaic redox reactions, where V_2_O_5_ and PmAP participated as electroactive materials. It can be noticed that charge–discharge rate remains uncharged when the current density is increased. The coulombic efficiency (η) of the V_2_O_5_/amino-FG/PmAP electrode was calculated to be around 92%, indicating its excellent cycling stability and rate capability. The specific capacitances of V_2_O_5_/amino-FG/PmAP and V_2_O_5_/amino-FG/PVDF electrodes as function of current density are illustrated in Figure 4e. The V_2_O_5_/amino-FG/PmAP electrode exhibited a remarkable *C_sp_* of 1438.4 F/g at 0.5 A/g, which is more than four times as low as that of V_2_O_5_/amino-FG/PVDF electrode (292.9 F/g). The greater *C_sp_* values of the V_2_O_5_/amino-FG/PmAP electrode could be ascribed to the higher level of pseudocapacitive contribution from V_2_O_5_ and PmAP components, which comprise of numerous redox active functionalities to enhance electron-transport ability of the nanocomposite electrode. In contrast, the non-conductive PVDF binder significantly suppressed electron exchange between active material and electrolyte during cycling. As the current density raised to 5 F/g, the *C_sp_* value decreased from 1438.4 F/g to 760.3 F/g. The V_2_O_5_/amino-FG/PmAP electrode can retain 76% and 53% of its capacitance after 4-fold and 10-fold increases in current density, respectively, indicating its good rate capability. The V_2_O_5_/amino-FG/PmAP electrode can deliver a maximum areal capacitance of 719.2 F/g and a minimum areal capacitance of 380.1 F/g at current densities of 0.5 and 5 A/g, respectively. In Figure 4f, a gradual decrease in the *C_sp_* of V_2_O_5_/amino-FG/PmAP electrode with increasing the cycle number is observed. However, the V_2_O_5_/amino-FG/PmAP electrode can retain its specific capacitance of 92% after 7000 cycles, which implies its impressive cycling life in KCl electrolyte.

The electrochemical performance of the V_2_O_5_/amino-FG/PmAP electrode was evaluated in Li_2_SO_4_ electrolyte and compared with KOH electrolyte. Figure 5a illustrates the CV profiles of V_2_O_5_/amino-FG/PmAP electrode in 1 M Li_2_SO_4_ at different scan rates. In the CV curves, a pair of anodic peaks can be found at 0.27 V and 0.38 V with a pair of cathodic peaks at 0.16 V and 0.31 V in the reverse scan. It is believed that the pseudocapacitive properties are caused by the redox interaction between V_2_O_5_/PmAP and Li_2_SO_4_. The current level of the CV curves gradually rises as scan rate increases. The nanocomposite electrode in 1 M Li_2_SO_4_ exhibits a lower current value and a smaller CV area than those in 1 M KCl. This might be due to the higher ionic radii of the hydrated SO_4_^2−^ ions (3.79 Å) and subsequently its lower ionic mobility than those of the hydrated Cl^−^ ions (3.32 Å). Hence, the insertion and desertion of SO_4_^2−^ ions into and out of the electrode matrix are much more difficult during cycling even though the ionic conductivity of sulfate ions is superior to that of chloride ions [58]. It can be concluded that the rate of diffusion of electrolyte ions plays an influential role in achieving better performance of the supercapacitors in addition to their ionic conductivity. Figure 5b shows the plots of faradaic peak currents against the scan rate at the logarithmic scale. The anodic and cathodic slope values are determined to be 0.725 and 0.683, respectively, which indicates that the capacitance performance is controlled by both diffusion and pseudocapacitance processes. The GCD profiles of V_2_O_5_/amino-FG/PmAP in 1 M Li_2_SO_4_ are demonstrated in Figure 5c. With increasing current density, discharge time gets shorter. The specific capacitances of V_2_O_5_/amino-FG/PmAP and V_2_O_5_/amino-FG/PVDF electrodes as a function of current density are presented in Figure 5d. The nanocomposites containing PmAP binder delivered significantly higher specific capacitance compared to those containing PVDF binder. The V_2_O_5_/amino-FG/PmAP electrode achieved a highest specific capacitance of 1191.7 F/g at 0.5 A/g in 1 M Li_2_SO_4_, while the capacitance value decreased to 510.3 F/g at 5 A/g. The V_2_O_5_/amino-FG/PmAP electrode showed 42% capacitance retention after 10-fold increases in current density, which is lower than that observed in KCl electrolyte (53% retention). The lower specific capacitance in Li_2_SO_4_ electrolyte could be ascribed to the higher volume of hydrated SO_4_^2−^ ions than Cl^−^, which sluggish the transport rate of SO_4_^2−^ ions and consequently enhance the internal resistance. The V_2_O_5_/amino-FG/PmAP electrode displayed 97% capacitance retention over 7000 GCD cycles in 1 M Li_2_SO_4_. The superior cycling stability in Li_2_SO_4_ is probably due to the strong solvation of Li^+^ and SO_4_^2−^ ions in the aqueous medium due to their high charge density. The Nyquist plots of V_2_O_5_/amino-FG/PmAP and V_2_O_5_/amino-FG/PVDF electrodes in different electrolytes are illustrated in Figure 5f. The impedance curves of V_2_O_5_/amino-FG/PmAP displayed a semi-circular arc in a high frequency regime together with a Warburg line in a low frequency regime. The V_2_O_5_/amino-FG/PmAP electrode reveals a smaller semicircle arc in KCl electrolyte, indicating its lower charge transfer resistance (*R_ct_*) than in Li_2_SO_4_ electrolyte. The *R_ct_* values of V_2_O_5_/amino-FG/PmAP electrode in KCl and Li_2_SO_4_ electrolytes are 46.2 Ω and 67.5 Ω, respectively. In contrast to Li_2_SO_4_, the smaller volume of hydrated K^+^/Cl^−^ ions is responsible for the lower *R_ct_* value in KCl. Furthermore, in aqueous 1 M KCl and 1 M Li_2_SO_4_, the nanocomposite electrode displays a solution resistance of 1.35 and 4.34, respectively. These results correlated well with the charge–discharge performances of the V_2_O_5_/amino-FG/PmAP electrode in these electrolytes. Furthermore, a steeper Warburg slope in KCl medium implies easier ionic movement across the aqueous KCl solution as well as the electrode–electrolyte interface. In contrast to the V_2_O_5_/amino-FG/PmAP electrode, the V_2_O_5_/amino-FG/PmAP electrodes possesses a smaller semicircle arc, which indicates their higher charge transfer resistance because PVDF is a poor electrical conductor than PmAP. The *R_ct_* values of V_2_O_5_/amino-FG/PmAP and V_2_O_5_/amino-FG/PVDF electrodes in 1 M KCl are 46.2 Ω and 257.9 Ω, respectively, which are well in agreement with their electrochemical performances.

Figure 6a demonstrates the CV profiles of solid-state V_2_O_5_/amino-FG/PmAP//activated carbon cloth asymmetric cell in 1 M KCl over various operating voltage windows at 25 mV/s. The CV results indicate that the as-assembled ASC cell is capable of stable operation up to as high as 1.4 V, which is further confirmed by the charge–discharge results (Figure 6c). As a result of extending the working voltage window, the integral CV area increases without changing their shape significantly, demonstrating a stable cyclic performance with a reasonable rate capacity of the ASC device when applied to a higher voltage window. The CV profiles are almost rectangular and have broad faradaic peaks, indicating that the ASC cell stored energy via both EDLC and pseudocapacitance processes. The CV curves of the ASC at different scan rates over a cell potential limit of 0–1.4 V are demonstrated in Figure 6b. With an increasing scan rate, the integral CV area increased gradually, indicating good rate capability. Figure 6d shows the GCD curves of as-assembled ASC cell at varying current densities, while Figure 6e illustrates the variation of specific capacitances as a function of current density. The maximum *C_sp_* value for the ASC could reach to 251.4 F/g at 0.5 A/g in KCl electrolyte. The ASC in KCl electrolyte also displayed reasonable capacitance retention, i.e., 54%, when current density was increased to 5 A/g. The ASC retained 93.2% capacitance after 4000 cycles in KCl electrolyte, implying its good cycling performance. The electrochemical performances of ASC cell in Li_2_SO_4_ electrolyte were evaluated and compared with those obtained with KCl electrolyte. Figure 7a illustrates the CV curves of ASC cells in Li_2_SO_4_ electrolyte at varying voltage ranges at 25 mV/s. As shown in Figure 7a, the ASC cell in Li_2_SO_4_ electrolyte can stably operate up to a working potential window of 0–1.7 V. Moreover, the CV results corroborated well by GCD results (Figure 7c). The excellent electrochemical stability in neutral Li_2_SO_4_ electrolyte over a wide potential window can be attributed to the low parasitic side reactions [5]. The asymmetrical rectangular shape of the CV curves indicates that both faradaic and non-faradaic mechanisms are involved in the charge storage process. The CV profiles of Li_2_SO_4_-based ASC device at different scan rates are shown in Figure 7b. There is no change in the shape of the CV curves as the scan rate increases, but the total CV area gradually increases. It can be noticed that at any given scan rate, the current level of the Li_2_SO_4_-based ASC is relatively lower than that found for KCl-based ASC, indicating better capacitive response of the ASC cell in KCl electrolyte. The GCD curves of ASC cell in Li_2_SO_4_ at different current densities are illustrated in Figure 7d. The asymmetric feature of the GCD profiles is further implies the occurrence of faradaic redox reaction during charge–discharge process. The specific capacitances of ASC cell in Li_2_SO_4_ electrolyte at different current densities are presented in Figure 7e. A maximum specific capacitance of 216.4 F/g at a current density of 0.5 A/g was achieved for the ASC cell in Li_2_SO_4_ electrolyte. The capacitance was reduced to 125.3 F/g at a current density of 5 A/g, which correspond to a 58% capacitance retention. The ASC with Li_2_SO_4_ electrolyte possesses outstanding cycling stability with 96.7% capacity retention after 4000 cycles compared to 93.2% retention with KCl electrolyte. The coulombic efficiency (η) of ASC in Li_2_SO_4_ electrolyte was calculated to be 83%, which indicates its excellent rate capability and electrochemical stability.

The impedance analyses of the ASC cell in both KCl and Li_2_SO_4_ electrolytes were conducted, and the resulting Nyquist plots are demonstrated in Figure 8a. The Nyquist plots of ASC cells exhibit a semicircular arc with Warburg straight line in the high and low frequency regions, respectively. The *R_ct_* values of ASC cells in KCl and Li_2_SO_4_ electrolytes are 51.7 Ω and 90.1 Ω, respectively. In contrast to Li_2_SO_4_ based ASC cell, the ASC cell based on KOH electrolyte exhibit steeper Warburg line, indicating faster ion transport. This could be attributed to the smaller volume of hydrated K^+^ and Cl^−^ ions than those of Li^+^ and SO_4_^2−^ ions. The Ragone plots of as-assembled ASC devices are demonstrated in Figure 8b. ASC device with Li_2_SO_4_ electrolyte delivered a highest energy density of 51.6 Wh/kg at a power density of 500 W kg^−1^, which significantly higher than those of KCl-based ASC cell (48.1 Wh/kg). The higher energy density of Li_2_SO_4_-based ASC cell is associated with the wider potential window (0–1.7 V). Furthermore, the ASC cell with Li_2_SO_4_ electrolyte retained the energy density of 29.6 Wh/ kg even at high power density of 6655 W/kg, while in contrast, the KCl-based ASC device can deliver the energy density of 26 Wh/kg at a power density of 4607 W/kg. Table 1 and Figure 8b illustrate a comparison of energy density at corresponding power density between present V_2_O_5_/amino-FG/PmAP//activated carbon ASC cell and previously reported ASC cells consisting of various ternary metal oxide/graphene/conducting polymer nanocomposites, which indicates better electrochemical performances of the present ASC cell compared to reported devices. Inset of Figure 8b displays the lightening of LED bulb using KCl-based ASC cell. The CV analysis of the as-assembled ASC cell in KCl electrolyte was carried out at different bending angles to evaluate its bending stability. Figure 8c illustrates that both shape and current level of the CV curves remain unchanged after bending the ASC cell from 0° to 135°, indicating its excellent bending stability.

## 4. Conclusions

In this study, we have demonstrated that PmAP can be potentially used as a conductive binder-cum-pseudocapacitive material instead of PVDF binder to fabricate self-standing supercapacitor electrodes with better capacitance performances. The electrochemical properties of as-prepared V_2_O_5_/amino-FG/PmAP nanocomposites were compared with those of V_2_O_5_/amino-FG nanocomposites with PVDF binder. The V_2_O_5_/amino-FG electrode with PmAP binder had a 5-fold higher specific capacitance in 1 M KCl electrolyte than the electrode with PVDF binder, which could be attributed to the higher electrical conductivity and pseudocapacitive properties of PmAP. EIS results revealed that the V_2_O_5_/amino-FG/PmAP electrode has much lower charge transfer resistance (46.2 Ω) than the electrode using PVDF as a binder (257.9 Ω). The type of electrolyte plays a crucial role in improving the capacitance of the V_2_O_5_/amino-FG/PmAP electrode. The specific capacitance is as high as 1438.4 F/g in 1 M KCl electrolyte and 1191.7 F/g in 1 M Li_2_SO_4_ electrolyte at 0.5 A/g. An as-assembled solid-state V_2_O_5_/amino-FG/PmAP//activated carbon cloth asymmetric cell exhibited higher energy density and better cycling stability with KCl electrolyte than with Li_2_SO_4_ electrolyte. The Li_2_SO_4_-based ASC cell can deliver maximum energy and power densities of 51.6 Wh/kg and 6655 W/kg, respectively, which are substantially higher than those of ASC devices constructed with various previously reported ternary composites as the positive electrode. We believe that PmAP could be used as a cheap binder-cum-pseudocapacitive material for next-generation supercapacitors with high energy density, compared with the PVDF binder currently used in commercial supercapacitors.

## Figures and Tables

**Figure 1 nanomaterials-13-00642-f001:**
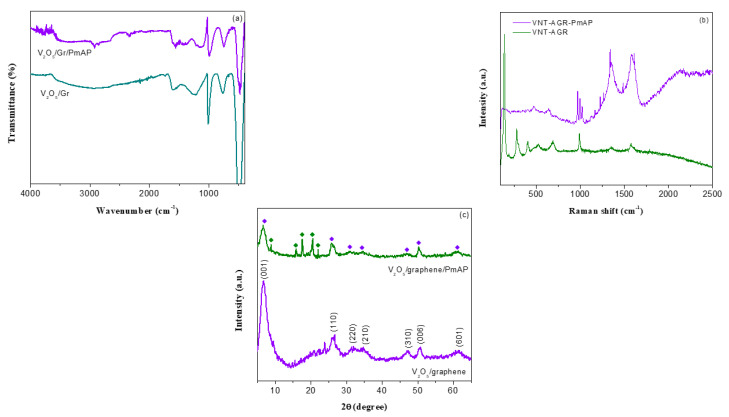
(**a**) FTIR spectra, (**b**) Raman spectra, and (**c**) XRD patterns of V_2_O_5_/amino-FG nanocomposite and V_2_O_5_/amino-FG/PmAP nanocomposite film.

**Figure 2 nanomaterials-13-00642-f002:**
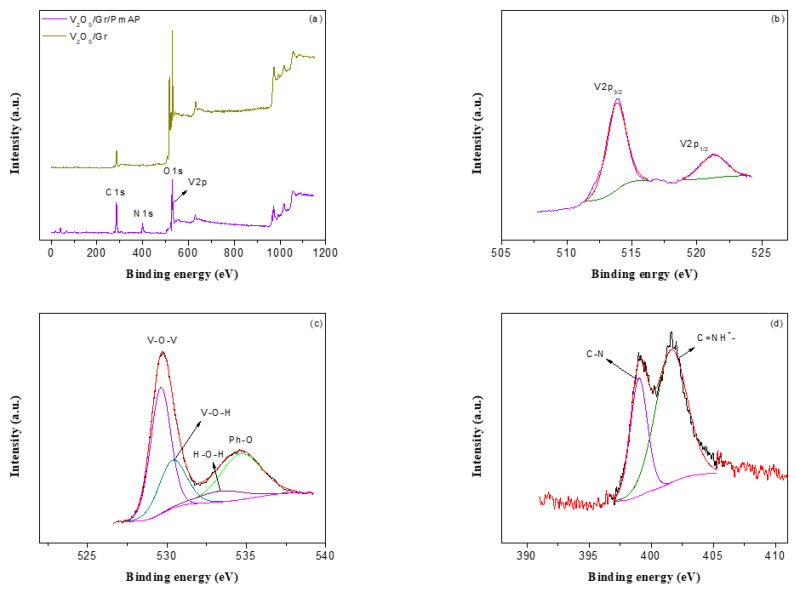
(**a**) XPS survey spectra XPS spectra of V_2_O_5_/amino-FG/PmAP nanocomposite film; the high-resolution spectra of (**b**) V 2p, (**c**) O 1s, and (**d**) N 1s, along with the fitting peaks of as-prepared V_2_O_5_/amino-FG/PmAP nanocomposite film.

**Figure 3 nanomaterials-13-00642-f003:**
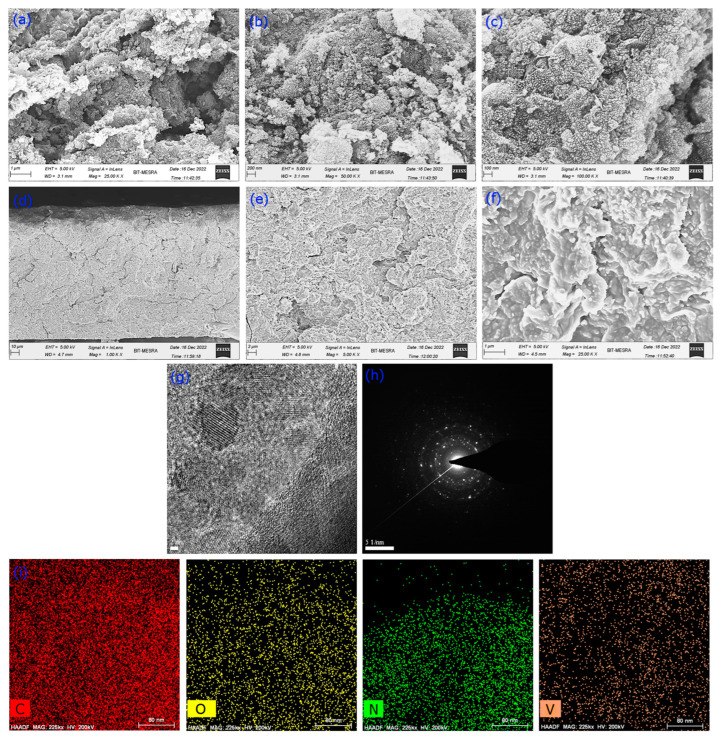
FESEM images of binary V_2_O_5_/amino-FG nanocomposite (**a**–**c**) and V_2_O_5_/amino-FG/PmAP nanocomposite film (**d**–**f**) under different magnifications. HR-TEM images of V_2_O_5_/amino-FG/PmAP nanocomposite film (**g**), Selected area electron diffraction (SAED) pattern (**h**), and EDX elemental mapping of V_2_O_5_/amino-FG/PmAP nanocomposite (**i**).

**Figure 4 nanomaterials-13-00642-f004:**
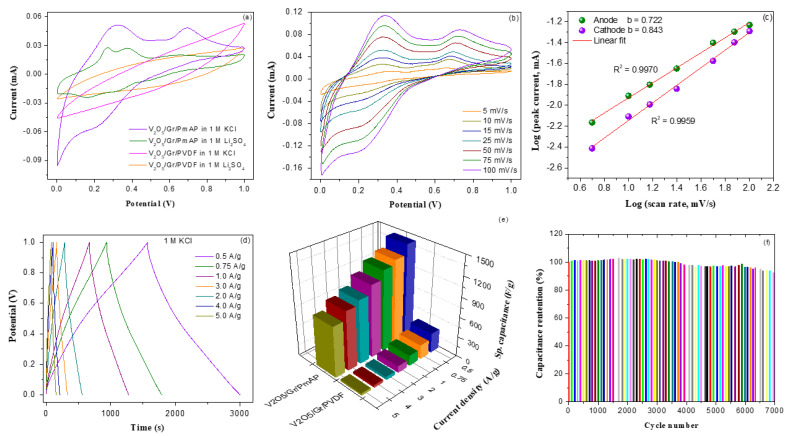
(**a**) CV curves of V_2_O_5_/amino-FG/PmAP and V_2_O_5_/amino-FG/PVDF nanocomposite in 1 M KCl and 1 M Li_2_SO_4_ at scan rate of 25 mV/s, (**b**) CV curves of V_2_O_5_/amino-FG/PmAP electrode in 1 M KCl electrolyte at different sweep rates, (**c**) Linear fitting curves between ln (i) Vs. ln (sweep rate), (**d**) Charge–discharge profiles of V_2_O_5_/amino-FG/PmAP nanocomposite at different current densities, (**e**) Variations of specific capacitance as function of current density for V_2_O_5_/amino-FG/PmAP and V_2_O_5_/amino-FG/PVDF nanocomposites film, and (**f**) Specific capacitance retention versus cyclic number for V_2_O_5_/amino-FG/PmAP electrode.

**Figure 5 nanomaterials-13-00642-f005:**
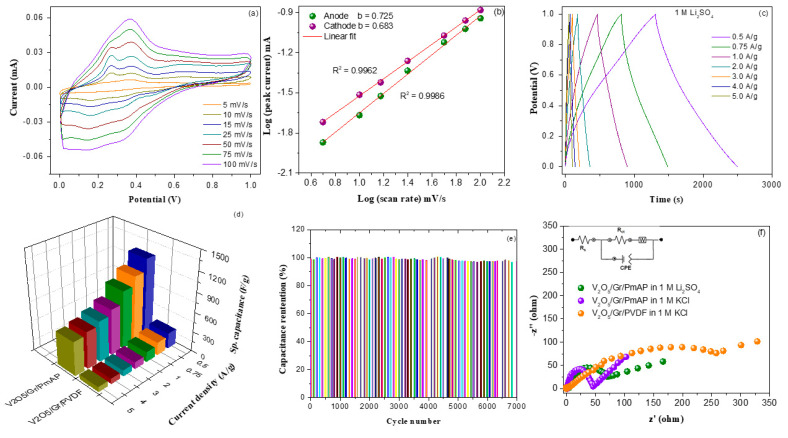
(**a**) CV curves of V_2_O_5_/amino-FG/PmAP electrode in 1 M Li_2_SO_4_ electrolyte at different sweep rates, (**b**) Linear fitting curves between ln (i) Vs. ln (sweep rate), (**c**) Charge–discharge profiles of V_2_O_5_/amino-FG/PVDF nanocomposites film in 1 M Li_2_SO_4_ electrolyte at different current densities, (**d**) Variations of specific capacitance as function of current density for V_2_O_5_/amino-FG/PmAP and V_2_O_5_/amino-FG/PVDF nanocomposites film, (**e**) Specific capacitance retention versus cyclic number for V_2_O_5_/amino-FG/PmAP electrode, and (**f**) EIS curves of V_2_O_5_/amino-FG/PmAP with 1 M KCl and 1 M Li_2_SO_4_ electrolyte and V_2_O_5_/amino-FG/PVDF electrodes in 1 M KCl with the equivalent electrical circuit model.

**Figure 6 nanomaterials-13-00642-f006:**
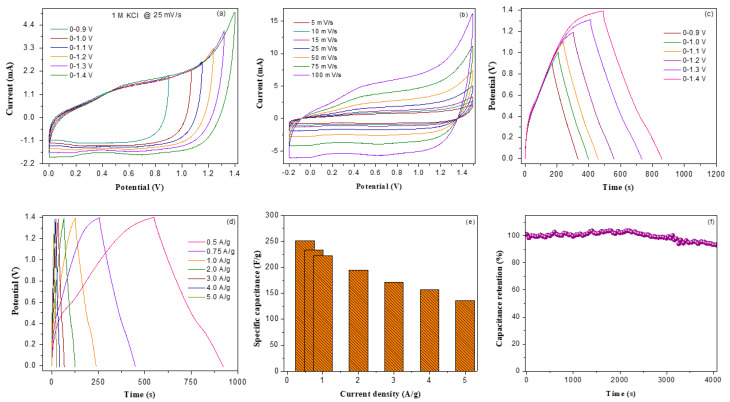
(**a**) CV curves of V_2_O_5_/amino-FG/PmAP//CC ASC device in 1 M KCl electrolyte with different potential windows ranging from 0–0.9 V to 0–1.4 V, (**b**) CV curves of the ASC device at different sweep rates, (**c**) Charge–discharge profiles of ASC device in 1 M KCl electrolyte with different potential windows ranging from 0–0.9 V to 0–1.4 V, (**d**) Charge–discharge (CD) cycling curves for the ASC device at different current densities, (**e**) Specific capacitance of ASC device over a range of current densities, and (**f**) Capacitance retention versus cyclic number for the ASC device in 1 M KCl.

**Figure 7 nanomaterials-13-00642-f007:**
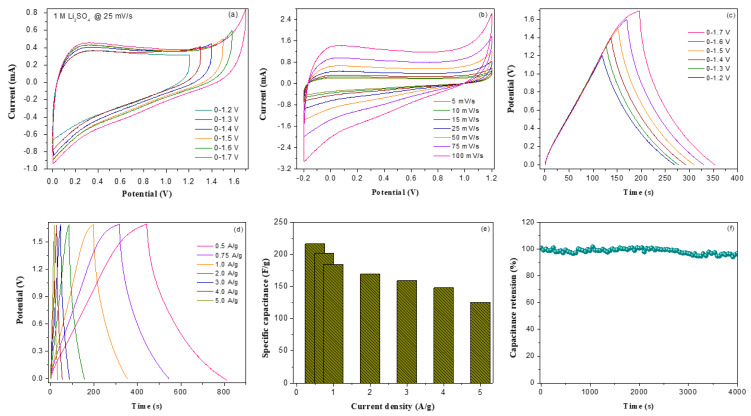
(**a**) CV curves of V_2_O_5_/amino-FG/PmAP//CC ASC device in 1 M Li_2_SO_4_ electrolyte with different potential windows ranging from 0–1.2 V to 0–1.7 V, (**b**) CV curves of the ASC device with 1 M Li_2_SO_4_ at different sweep rates, (**c**) Charge–discharge profiles of ASC device in 1 M Li_2_SO_4_ with different potential windows, (**d**) Charge–discharge (CD) cycling curves for the ASC device at different current densities, (**e**) Specific capacitance of ASC device over a range of current densities, and (**f**) Capacitance retention versus cyclic number for the ASC device in 1 M Li_2_SO_4_.

**Figure 8 nanomaterials-13-00642-f008:**
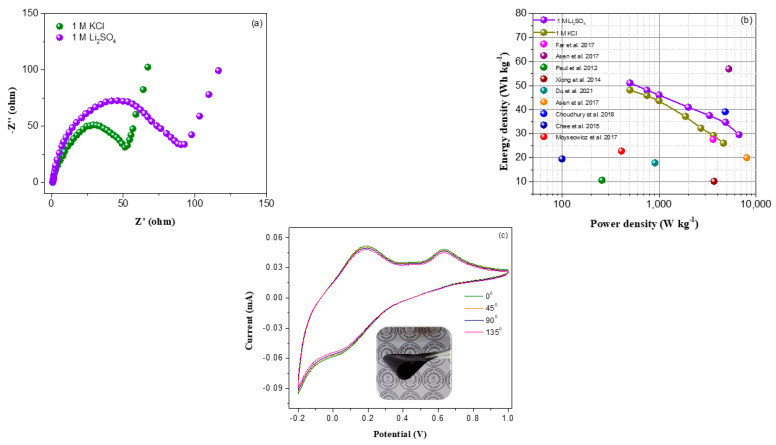
(**a**) Nyquist plots of V_2_O_5_/amino-FG/PmAP//CC ASC device in 1 M KCl and 1 M Li_2_SO_4_, (**b**) Ragone plots of the ASC devices in 1 M KCl and 1 M Li_2_SO_4_ and earlier reported similar types of ASC devices [24,61,62,63,64,65,66,67,68], and (**c**) CV profiles of as-assembled asymmetric device in KCl electrolyte after bending at different angles.

**Table 1 nanomaterials-13-00642-t001:** Comparison of the reported ASCs with recent devices using asymmetric configuration design.

ASCs Cell Configuration		Electrolyte	ED (Wh/kg)	PD (W/kg)	Ref.
Anode	Cathode				
NiO-CoO-PPy	AC	2 M KOH	36	801	[69]
GO/PPy	AC	1 M Na_2_SO_4_	21.4	453.9	[70]
PANI/Ag@MnO_2_	AC	2 M KOH	49.77	1599.75	[71]
MnO_2_@CNF	ACNF	1 M Na_2_SO_4_	8.7	2080	[72]
PANI–MnO_2_	AC	6 M KOH	20	400	[73]
PPy/FeCoS-rGO	rGO	3 M KOH	28.3	810	[74]
Co_3_O_4_/PPy/MnO_2_	AC	1 M NaOH	34.3	80	[75]
Gra/PEDOT/MnO_2_	AC	0.5 M Na_2_SO_4_	31.4	90	[76]
V_2_O_5_ NF	PANI NF	3 M KCl	26.7	220	[77]
V_2_O_5_/amino-FG/PmAP	AC	1 M KCl	26	4607	This work
V_2_O_5_/amino-FG/PmAP	AC	1 M Li_2_SO_4_	29.6	6655	This work

## Data Availability

Not applicable.

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
