# Peer review of "Fabrication of Flexible Poly(m-aminophenol)/Vanadium Pentoxide/Graphene Ternary Nanocomposite Film as a Positive Electrode for Solid-State Asymmetric Supercapacitors"

_nanomaterials, 2023, doi:10.3390/nano13040642_

Round 1
Reviewer 1 Report
Asymmetric supercapacitors are superior to symmetric supercapacitors in energy density and stability. The topic of this manuscript is interesting. However, major revisions are required and the comments are given below.
1. The authors need to provide more discussion on the innovation and significance of this work. Importantly, the authors should provide more discussion on the future perspectives of the presented research in the abstract, introduction, and conclusion to sublimate manuscript.
2. Various electrode materials including porous carbon, conducting polymers transition metal compounds have been developed for supercapacitors. More typical references are suggested to be cited for broad readers, e.g. Journal of Bioresources and Bioproducts 2021, 6 (2), 142-151; Journal of Bioresources and Bioproducts 2022, 7 (1), 63-72; Inorganic Chemistry Frontiers 2022, 9, 6108-6123.
3. “graphene, a thick 2D carbon sheet” in line 53 should be revised as “graphene, a thin 2D carbon sheet”. Please go through the whole manuscript to remove such incorrect expressions.
4. Please pay attention to the writing of units. For example, the units should be written in the same style, “F/g” and “F g-1” need to be revised.
5. The authors should mention where did they get or how did they synthesis activated carbon cloth.
6. For V2O5/amino-FG/PmAP nanocomposite electrodes, 250 mg PmAP was mixed with 1 g V2O5/amino-FG, the mass ratio of binder (PmAP) is 20%. While the mass ratio of PVDF is 15% which is not suggested to be compared with 20% PmAP.
7. For Nyquist plots, the scale of X axial and Y axial is required to be the same. Please refer to New Journal of Chemistry 2022, 46, 10844-10853.
8. How about the capacitance performance of activated carbon cloth?
9. How about the capacitance performance of ASC compared to those assembled by other activated materials?
Author Response
Answers to Reviewer’s Comments on the Ms. Ref. No.: nanomaterials-2196721R1
The following are the answers to the comments of the reviewers on our paper entitled “Fabrication of flexible poly(m-aminophenol)/vanadium pentoxide/graphene ternary nanocomposite film as a positive electrode for solid-state asymmetric supercapacitors”. We have made necessary modifications to the revised MSS in light of the reviewer's comments. The changes included in the revised MSS are highlighted in red color.
|
Reviewer’s comments |
Answer’s to the comments |
|
Reviewer #1
|
|
|
The authors need to provide more discussion on the innovation and significance of this work. Importantly, the authors should provide more discussion on the future perspectives of the presented research in the abstract, introduction, and conclusion to sublimate manuscript. |
As suggested by reviewer, we have included the advantages of using PmAP as polymeric binder over other commonly used polymer binders for fabrication of supercapacitor electrodes in all three sections in the revised MSS. In the present investigation, we have used PmAP as a conductive binder-cum-pseudocapacitive material, which has been highlighted in all three sections as mentioned by reviewer. |
|
Various electrode materials including porous carbon, conducting polymers transition metal compounds have been developed for supercapacitors. More typical references are suggested to be cited for broad readers, e.g. Journal of Bioresources and Bioproducts 2021, 6 (2), 142-151; Journal of Bioresources and Bioproducts 2022, 7 (1), 63-72; Inorganic Chemistry Frontiers 2022, 9, 6108-6123. |
We have included all suggested references in the introduction section with the reference number of 3, 9, 10, in the revised MSS. |
|
“graphene, a thick 2D carbon sheet” in line 53 should be revised as “graphene, a thin 2D carbon sheet”. Please go through the whole manuscript to remove such incorrect expressions. |
As per reviewer comment, we have made the necessary corrections in the revised MSS. |
|
Please pay attention to the writing of units. For example, the units should be written in the same style, “F/g” and “F g-1” need to be revised. |
As per reviewer comment, we have revised all the units in same style in the revised MSS. |
|
The authors should mention where did they get or how did they synthesis activated carbon cloth. |
We have directly purchased the activated carbon cloth from Sigma-Aldrich, India, which is suggested to mentioned in the revised MSS. |
|
For V2O5/amino-FG/PmAP nanocomposite electrodes, 250 mg PmAP was mixed with 1 g V2O5/amino-FG, the mass ratio of binder (PmAP) is 20%. While the mass ratio of PVDF is 15% which is not suggested to be compared with 20% PmAP. |
Thank you for your comment. It was a writing mistake. We have used same amount of PmAP and PVDF that is 20 Wt%. |
|
For Nyquist plots, the scale of X axial and Y axial is required to be the same. Please refer to New Journal of Chemistry 2022, 46, 10844-10853. |
As per reference suggested by reviewer, we have modified the scales of both axis in Nyquist plot (Figure 5f and 8a). |
|
How about the capacitance performance of activated carbon cloth? |
We have already conducted the CV analysis of this activated carbon cloth whatever used in the present investigation, and the results are reported in our earlier paper published in Anand et al. Nanotechnology 32 (2021) 495403. |
|
How about the capacitance performance of ASC compared to those assembled by other activated materials? |
The comparison among the capacitance performance of ASC cell with other reported electrode materials is included in revised Table 1. |
Reviewer 2 Report
The authors reported a novel flexible poly(m-aminophenol)/vanadium pentoxide/graphene ternary nanocomposite film as a positive electrode for solid-state asymmetric supercapacitors. The introduction of poly(m-aminophenol) plays an important role in increasing the electrochemical properties. The rich data well supports their point of view. So I recommend its publication in nanomaterials after clarifying the following problems:
1) The use of poly(m-aminophenol) is claimed to be key for improving the electrochemical properties of electrode materials. However, the electrochemical property of poly(m-aminophenol) is not studied in detail. In previous studies, the poly(m-aminophenol) is found to possess pseudo-capacitive activity. So the role of poly(m-aminophenol) in this material system should be clarified in detail.
2) Considering the key role of poly(m-aminophenol), a review of its application in supercapacitors should be added in the introduction part, which can help readers quickly grasp its advances.
3) The V2O5/amino-FG/PVDF electrode is chosen for comparison in this study. Why is the hydrophobic PVDF chosen as the binder rather than the hydrophilic PTFE in the aqueous electrolyte?
4) In the part of Electrochemical Measurements, the electrode mass should be provided.
5) In line 283-285, the description is confusing, because the authors evaluated the reversibility based on the peak separation potential rather than the peak current.
6) In line 307-308, the capacitance behavior is attributed to the high ionic mobility of the Cl- ions. That is also confusing, because the reaction mechanism of the V2O5 for energy storage should be related to the Insertion and deinsertion of K+. Thus the energy-storage mechanism is suggested to add in the manuscript.
7) For figure 4a, the CV curves of the V2O5/Gr/PmAP in KCl and Li2SO4 show different peaks, what is the reason for that? The underlying mechanism should be discussed.
8) As a flexible electrode material for supercapacitors, the authors studied its properties under different bending states. In that aspect, the latest advances and requirements of flexible and wearable supercapacitor electrodes should be cited and discussed, such as doi: 10.1016/j.matt.2021.07.021; 10.1007/s42765-022-00162-7
9) Some grammatical errors should be corrected, such as line 22, electrolyte should be electrode. Please check the manuscript and correct them.
Author Response
Answers to Reviewer’s Comments on the Ms. Ref. No.: nanomaterials-2196721R1
The following are the answers to the comments of the reviewers on our paper entitled “Fabrication of flexible poly(m-aminophenol)/vanadium pentoxide/graphene ternary nanocomposite film as a positive electrode for solid-state asymmetric supercapacitors”. We have made necessary modifications to the revised MSS in light of the reviewer's comments. The changes included in the revised MSS are highlighted in red color.
|
Reviewer’s comments |
Answer’s to the comments |
|
Reviewer #2 |
|
|
The use of poly(m-aminophenol) is claimed to be key for improving the electrochemical properties of electrode materials. However, the electrochemical property of poly(m-aminophenol) is not studied in detail. In previous studies, the poly(m-aminophenol) is found to possess pseudo-capacitive activity. So the role of poly(m-aminophenol) in this material system should be clarified in detail. |
As suggested by reviewer, we have included the importance of using soluble conducting polymer like PmAP as a conductive binder for fabrication of supercapacitor electrode in the revised MSS. In our earlier work, we have explored PmAP as a pseudocapacitive material to develop nanocomposite electrode with carbon nanofibers (CNFs). We have observed a significant enhancement of specific capacitance of CNFs upon inclusion of PmAP as pseudocapacitive component. Based of the electrochemical performances of PmAP as observed our earlier work, we have used it as a conductive binder-cum-pseudocapacitive material for fabrication of ternary nanocomposite-based electrode for supercapacitor applications in the present investigation. We have cited our previous work as a reference number 22 in the revised MSS |
|
Considering the key role of poly(m-aminophenol), a review of its application in supercapacitors should be added in the introduction part, which can help readers quickly grasp its advances. |
In the present investigation, we have used PmAP as a conductive binder-cum-pseudocapacitive material, which has been highlighted in all three sections as mentioned by reviewer. |
|
The V2O5/amino-FG/PVDF electrode is chosen for comparison in this study. Why is the hydrophobic PVDF chosen as the binder rather than the hydrophilic PTFE in the aqueous electrolyte? |
There are some important reasons behind selection PVDF as a binder for comparison.
|
|
In the part of Electrochemical Measurements, the electrode mass should be provided. |
As suggested by reviewer, we have mentioned the total mass of the hybrid electrode, i.e., 3.417 mg, in the experimental section of the revised MSS. |
|
In line 283-285, the description is confusing, because the authors evaluated the reversibility based on the peak separation potential rather than the peak current. |
We have removed the confusion by deleting the sentence. |
|
In line 307-308, the capacitance behavior is attributed to the high ionic mobility of the Cl- ions. That is also confusing, because the reaction mechanism of the V2O5 for energy storage should be related to the Insertion and deinsertion of K+. Thus the energy-storage mechanism is suggested to add in the manuscript. |
We are sorry for the error. Accordingly, we have modified the sentence as “probably because of high ionic mobility of the K+ ions” in the revised MSS. The probable electrode reaction would be: V2O5 + nK+ + ne- ↔ KnV2O5 The possible reaction mechanism has been included in the revised MSS. |
|
For figure 4a, the CV curves of the V2O5/Gr/PmAP in KCl and Li2SO4 show different peaks, what is the reason for that? The underlying mechanism should be discussed. |
It is quite obvious that the shape and size of the CV profile are very much dependent on the nature of electrolytes. The different CV profiles for different electrolyte could be attribute to different kinetics of electrochemical reactions for different electrolytes. The diffusion kinetics of different electrolyte ions are different because of their different size/mobility, which results in different shape and size of CV curves. |
|
As a flexible electrode material for supercapacitors, the authors studied its properties under different bending states. In that aspect, the latest advances and requirements of flexible and wearable supercapacitor electrodes should be cited and discussed, such as doi: 10.1016/j.matt.2021.07.021; 10.1007/s42765-022-00162-7 |
We have included all suggested references in the introduction section with the reference number of 3, and 4 in the revised MSS. |
|
Some grammatical errors should be corrected, such as line 22, electrolyte should be electrode. Please check the manuscript and correct them. |
Sorry for the mistake. We have made the necessary corrections in the revised MSS. |

Round 2
Reviewer 1 Report
The manuscript has been revised according to the comments and suggest to be accepted.
Reviewer 2 Report
After revision, the problems have been well solved and the quality of the manuscript has been greatly improved. So I recommend its publication in Nanomaterials.